# The Contribution of Functional Magnetic Resonance Imaging to the Understanding of the Effects of Acute Physical Exercise on Cognition

**DOI:** 10.3390/brainsci10030175

**Published:** 2020-03-18

**Authors:** Fabian Herold, Norman Aye, Nico Lehmann, Marco Taubert, Notger G. Müller

**Affiliations:** 1Research Group Neuroprotection, German Center for Neurodegenerative Diseases (DZNE), Leipziger Str. 44, 39120 Magdeburg, Germany; notger.mueller@dzne.de; 2Department of Neurology, Medical Faculty, Otto von Guericke University, Leipziger Str. 44, 39120 Magdeburg, Germany; 3Institute III, Department of Sport Science, Otto von Guericke University Magdeburg, Zschokkestr. 32, 39104 Magdeburg, Germany; norman.aye@ovgu.de (N.A.); nico1.lehmann@ovgu.de (N.L.); marco.taubert@ovgu.de (M.T.); 4Center for Behavioral Brain Sciences (CBBS), Brenneckestraße 6, 39118 Magdeburg, Germany

**Keywords:** cognition, acute exercise, fMRI, physical activity breaks

## Abstract

The fact that a single bout of acute physical exercise has a positive impact on cognition is well-established in the literature, but the neural correlates that underlie these cognitive improvements are not well understood. Here, the use of neuroimaging techniques, such as functional magnetic resonance imaging (fMRI), offers great potential, which is just starting to be recognized. This review aims at providing an overview of those studies that used fMRI to investigate the effects of acute physical exercises on cerebral hemodynamics and cognition. To this end, a systematic literature survey was conducted by two independent reviewers across five electronic databases. The search returned 668 studies, of which 14 studies met the inclusion criteria and were analyzed in this systematic review. Although the findings of the reviewed studies suggest that acute physical exercise (e.g., cycling) leads to profound changes in functional brain activation, the small number of available studies and the great variability in the study protocols limits the conclusions that can be drawn with certainty. In order to overcome these limitations, new, more well-designed trials are needed that (i) use a more rigorous study design, (ii) apply more sophisticated filter methods in fMRI data analysis, (iii) describe the applied processing steps of fMRI data analysis in more detail, and (iv) provide a more precise exercise prescription.

## 1. Introduction

In Western societies, technological advances constantly lower the need to be physically active in every-day life situations (e.g., transportation to the workplace or physical demands during work) [1,2,3,4]. Hence, it is not surprising that the time spent physically inactive and sedentary has increased dramatically [5,6,7]. In the long term, rising levels of physical inactivity are expected to cause serious health problems [8]. For instance, sedentary behavior [9,10] and physical inactivity [11,12] are associated with worsened cognitive functions and a higher risk of neurodegenerative diseases (e.g., dementia) in the aging population [13,14,15,16,17,18,19,20,21,22]. In order to avoid periods of prolonged physical inactivity and sedentary behavior (e.g., sedentism during office working day), it is advised to take physical activity breaks ([23,24,25]. There is substantial evidence in the literature showing that acute (single) bouts of physical exercise (defined as planned and structured form of distinct physical activities [26,27,28,29]) improve cognitive domains, such as attention and/or cognitive control substantially, albeit only transiently [30,31,32,33]. However, the underlying neurobiological mechanisms of these effects are not yet fully understood. In this regard, the use of neuroimaging methods offers great potential for acquiring a deeper understanding of physical exercise-induced changes in the neural correlates of cognition, such as changes in functional brain activation [33,34,35]. The most common methods used to investigate effects on functional brain activation are functional near-infrared spectroscopy (fNIRS) [34] and electroencephalography (EEG) [36,37]; however, also functional magnetic resonance imaging (fMRI) has recently been applied in the context of acute physical exercise and cognition [38,39]. The strengths of fNIRS and EEG compared to fMRI are a higher temporal resolution, greater portability, and applicability in almost all cohorts (e.g., for individuals with metallic implants or claustrophobia) [34,40,41]. However, fNIRS and EEG have a limited spatial resolution and only allow for the evaluation of brain activation patterns in cortical areas [41,42,43]. In comparison to fNIRS and EEG, fMRI enables the assessment of brain activation changes in cortical and subcortical areas and offers a higher spatial resolution, which results in superior source localization [42,43,44]. Hence, fMRI is well suited to study the influence of acute physical exercise on subcortical structures, such as the hippocampus, which have a crucial role in cognitive processes (e.g., memory) [45,46,47,48,49]. Since hippocampal activation during cognitive tasks can distinguish between individuals with cognitive impairment and healthy controls [50,51,52], the investigation of the influence of acute physical exercise on these subcortical structures may help to develop more efficient physical intervention strategies with regard to neuroprotection. Furthermore, given the ability of fMRI to measure both subcortical and cortical activation, the application of fMRI prior to and after an acute bout of physical exercise is also useful to complement the results obtained by EEG and/or fNIRS.

fMRI brain (neural) activity is indirectly assessed by using the magnitude of venous blood oxygenation level-dependent contrast (BOLD) [53,54,55,56,57,58,59]. While BOLD contrast has been shown to reflect neural activity [60,61], this activity is also influenced by blood volume, oxygenation, and changes in regional blood flow [57,59,62]. An increase in neuronal activity (e.g., as a consequence of solving a cognitive task) induces higher neuronal energy metabolism, in which oxygen is consumed to produce energy in order to satisfy the energetic demands of neural tissue (neurometabolic coupling). This leads to a local decrease in the concentration of oxygenated hemoglobin and a local increase in the concentration of deoxygenated hemoglobin [58,63,64,65,66,67]. Furthermore, an increase in neural activity also stimulates local changes in cerebral hemodynamics, which trigger intensified blood flow to the activated brain regions (neurovascular coupling). An increase in the local cerebral blood flow leads to an increased washout of deoxygenated hemoglobin and ensures that an appropriate level of oxygen is supplied to the activated neuronal tissue [65,66,67,68,69]. A schematic illustration of these neurobiological mechanisms is provided in Figure 1.

Based on the paramagnetic properties of deoxyhemoglobin, neural activity-dependent changes in the concentration of deoxygenated hemoglobin influence the local reduction in main field homogeneity and, thus, the magnitude of the BOLD contrast [57,58,70]. Changes in the BOLD signal can be analyzed with different methods, such as task-based fMRI analyses (e.g., BOLD changes obtained during a cognitive task) or resting-state fMRI analyses (e.g., BOLD changes obtained while performing no explicit task). Recently, several studies using task-based fMRI analyses [38,71,72] and resting-state fMRI analyses [73] were performed to investigate the changes in neural correlates driving cognitive improvements after acute physical exercise. While there are systematic reviews available that provide recommendations with regard to acute exercise [33] or fMRI [62,74,75], no systematic review has pooled the methodology and findings of fMRI studies dealing with acute physical exercise and cognition. As the application of fMRI in the field of acute physical exercise and cognition poses specific challenges to the study design and analysis of fMRI signals (e.g., the influence of systemic artifacts, the influence of temporal window between exercise cessation, and an fMRI scan), previously published recommendations might not be fully transferable. Hence, this literature survey aims to systematically summarize the methodological details and findings of all studies using fMRI to investigate the influence of acute physical exercise on cognition. Based on the results of this systematic review, recommendations for future studies are derived.

## 2. Methods

In this systematic review, we followed the recommendations provided in the PRISMA statement (Preferred Reporting Items for Systematic Reviews and Meta-Analyses) [76]. Two independent researchers performed a systematic literature search in the following five electronic databases (applied specifications): Pubmed (all fields), Scopus (title, abstract, keywords), Web of Science (all fields), PsycInfo (all fields), and SportDiscus (all fields) on 3^rd^ March 2020. In this process, the following terms were used as search strings: 

Search strategy


“acute exercis*” OR “acute aerobic” OR “acute strength” OR “acute resistance” OR “acute effects of exercis*” OR “single bout of exercis*” OR “acute bout of physical activity” OR “single bout of physical activity” OR “acute physical activity” OR “physical activity break” 
AND 
mental OR neuropsychological OR brain OR cogniti* OR neurocogni* OR executive OR attention OR memory OR “response time” OR “reaction time” OR accuracy OR error OR inhibition OR visual OR spatial OR visuospatial OR processing OR recall OR learning OR language OR oddball OR “task switching” OR “problem solving” OR Flanker OR Stroop OR Sternberg 
AND 
fMRI OR MRI OR “MR imaging” OR “magnetic resonance imaging” 


Afterward, the results of the systematic search were loaded into a citation manager (Citavi 6.3), and the duplicates were removed (see Figure 2).

### 2.1. Inclusion and Exclusion Criteria

To screen for relevant studies, the established PICOS-principle was applied [76,77]. “PICOS” stands for participants (P), intervention (I), comparisons (C), outcomes (O), and study design (S) [76,77]. In the current systematic review, the following inclusion and exclusion criteria were used: (P) we applied no restrictions and included all age groups regardless of pathologies; (I) only acute physical exercise interventions were considered; (C) in this systematic literature search, no specific restrictions were used; (O) studies were considered relevant if they assessed changes in functional brain activation using fMRI (resting-state fMRI studies were included when they investigated the relationship between resting-state measures and cognitive performance). As cognition comprises a wide range of mental abilities that allow us, for example, to perceive, process, and retain information, as well as interact with our environment [78,79], it is necessary to classify the cognitive domains included in this systematic review. We included studies utilizing tests for cognitive domains such as attention, cognitive control, and memory [30,33] and excluded studies investigating solely perception (e.g., pain perception) or motor learning. (S) Only interventional studies that fulfill the previously stated criteria and have been published in a peer-reviewed journal were considered. 

As shown in Figure 2, 16 studies were excluded after full-text screening because they did not meet our inclusion criteria. In particular, six studies did not perform neuroimaging using fMRI [80,81,82] or applied other neuroimaging techniques (e.g., functional near-infrared spectroscopy) [83,84,85]. The remaining 10 studies did not perform cognitive testing [86,87,88,89,90,91,92,93,94] or did not report data because they describe a study protocol [95].

### 2.2. Data Extraction

We extracted information about the first author, year of publication, population characteristics including health status, cardiorespiratory fitness level, age, gender, and anthropometric data (e.g., body weight, body height, and body-mass index [BMI]). Furthermore, details about exercise characteristics (e.g., type of physical exercise, exercise intensity, and exercise duration), cognitive testing (e.g., tested cognitive domain and administration after exercise cessation), and (pre-)processing of fMRI data (e.g., filter cut-off frequencies) were extracted. Moreover, information about the main behavioral findings and neuroimaging findings were gathered from the studies considered to be relevant. In addition, we extracted the x-y-z coordinates of significant exercise-related functional hemodynamic changes (if possible) and performed automatic anatomical labeling using AAL2 [96] to obtain uniform neuroanatomical labeling across the reviewed studies. In three studies, extraction of the exact x-y-z coordinates was not possible [38,72,73].

### 2.3. Risk of Bias Assessment

As recommended in the PRISMA statement, the methodological quality of the included studies and their risk of bias was assessed (see Figure 3) [76]. In this process, two independent evaluators used the Cochrane Collaboration’s Risk of Bias tool and rated the risk of bias as being “low”, “high”, or “unclear” [97]. Any discrepancies between the two evaluators regarding the ratings of the risk of bias in a particular study were resolved by a discussion among them or through consultation with a third author of the review [97].

## 3. Results

### 3.1. Risk of Bias 

As shown in Figure 3, in the domains of sequence generation, allocation concealment, blinding of participants and personnel, and blinding of outcome assessments, all studies were rated as having an unclear risk of bias because the applied procedures were not described in sufficient detail. In the domains of incomplete outcome data and selective reporting, all studies were judged as having a low risk of bias. Furthermore, with regard to the domain of “other bias”, the majority of studies were judged as having an unclear bias because the study design did not meet recent recommendations [33] (e.g., it used only a posttest comparison [38,39,71,73,99,100,101,102,103,104] and/or the prescription of exercise intensity was not optimal because it was not based on a graded exercise test [39,71,72,73,103,104,105]).

### 3.2. Study Characteristics

In the reviewed studies, the acute effects of physical exercise were studied in healthy children [71,73], adolescents with and without bipolar disorder [105], younger adults with attention deficit hyperactivity disorder (ADHD) [100,102], healthy younger adults [38,39,72,99,101,102,106], and healthy older adults [103,104,107]. A detailed overview of the sample characteristics (e.g., age, gender, BMI, and cardiorespiratory fitness level) can be found in Table 1. The sample size in the reviewed studies varied between 9 [71,73,99] and 34 participants [107]. In particular, the following sample sizes were used: < 10 participants [71,73,99], between 10 to 20 participants [38,39,72,99,102,106], and > 20 participants [100,101,103,104,105,107].

Twelve of the studies were conducted in a within-subject design [38,39,71,72,73,100,101,102,103,104,106,107] and six studies used a between-subjects design [99,100,101,102,105,106]. Furthermore, in four studies, a cognitive test using fMRI was performed prior to and after cycling [72,105,106,107], whereas in 10 studies, testing was only performed after the cessation of the intervention conditions (e.g., cycling or seated rest/watching movie) [38,39,71,73,100,101,102,103,104,106]. In these 10 studies, the order of the separate testing sessions was counterbalanced [38,39,71,73,100,101,102,103,104,106]. Furthermore, the testing sessions were separated by a time interval of at least two days [38,100,101,102], seven days [39,71,73,106], or twelve days [103], and four studies reported that the testing sessions were performed at the same time of day [38,71,73,106]. After exercise cessation, the cognitive tests were administered based on the following criteria: (i) after 5 min [38], (ii) after 10 min [103], (iii) after approximately 15 min [72,104], (iv) within 15 min [39], (v) after 15.1 ± 4.3 min (moderate-intensity conditions) or 16.5 ± 5.1 min (light-intensity conditions) [107], (vi) after 20 min [106], (vii) 20.9 ± 1.8 min [100], (viii) after 25 min [105], (ix) approximately 63:38 min after exercise cessation [99], or (x) after reaching a level within 10% of the subject’s resting heart rate [71,73]. 

To test the cognitive performance of participants, in one study, a monetary incentive delay task [99], a mnemonic discrimination task [38], and a semantic memory task were implemented [104]. In two studies, a sustained attention task [72,105] or a checkerboard task [101,102] were applied. Four studies utilized the Flanker task [73,101,102,103] to test executive performance. In three studies a Go/No-Go task was used [100,101,102]. In four studies, working memory performance was examined by the N-Back paradigm [39,71,106,107]. In seven studies, cognitive testing was conducted using a block design [39,71,72,101,102,105,106], and in eight studies, an event-related design was applied [38,99,100,101,102,103,104,107]. 

### 3.3. Exercise Characteristics

All of the reviewed studies conducted an endurance exercise outside an fMRI scanner using a stationary cycling ergometer [38,39,71,72,73,100,101,102,103,104,105,106,107] or a treadmill [99]. In six studies, a graded exercise test was performed to determine the cardiorespiratory fitness level [38,99,100,101,102,107], and in one study, the cardiorespiratory fitness level was assessed via an indirect maximal oxygen uptake test [106]. The exercise intensity was set to HR 60–70% of the predicted maximal heart rate (HR_max_) [39,71,72,73], 60–80% of the predicted HR_max_ [105], 50–70% of the measured HR_max_ [100,101,102], 65% of the measured HR_max_ [107], >70% of the measured HR_max_ [101], 30% of the subject’s VO_2 peak_ [38], or 60–70% of the subject’s VO_2 max_ [99]. In two studies, the Borg scale was used as a subjective rating of the perceived exertion, which ranges from 6 (no exertion) to 20 (maximal exertion) [108]; a rating of 15 (hard) was used to set the exercise intensity [103,104]. In five studies, HR_max_ was predicted using the formula ‘220 – age in years’ [39,71,72,73,105], and one study used the Karvonen formula to calculate the target HR for cycling [106].

The exercise duration was set to 10 min [38], 20 min [39,72,103,104,105,106,107], 21 min [101], or 30 min [71,73,99,100,101,102]. The durations of the warm-up were set to 2 min [71,73] or 5 min [39,100,101,103,104,105,106], whereas the period of cool-downs lasted 2 min [105], 3 min [71,73], 4 min [101], or 5 min [39,103,104,106,107].

### 3.4. fMRI Characteristics and Data Processing

The majority of the studies used a 3.0 T MR scanner [38,39,71,72,73,100,101,102,103,104,105,106,107] equipped with 8 channel head coils [72,105], 12 channel head coils [100,101,102,107], 32 channel head coils [38,39,103,104,106,107], or 64 channel head coils [100,101,102]. To process the data, SPM 8 (statistical parametric mapping) [39,99], SPM 12 [100,101,102,106], DPABI (a toolbox for Data Processing and Analysis for Brain Imaging) [71,73], FSL (Functional MRI of the Brain Software Library) [72,105,107], or AFNI (Analysis of Functional NeuroImages) were applied [38,103,104,107]. Using these software packages, the reviewed studies commonly conducted the following “basic” processing steps: (i) rejection of the first 3 [71,105], first 5 [39,104], first 6 [103], first 10 [73], or first 12 images [106]; (ii) slice-timing correction [38,39,71,73,99,103,106]; (iii) realignment (motion correction) [39,71,73,99,100,101,102,103,105,106]; (iv) intra-subject and inter-modal registration (e.g., to T1-weighted image) and spatial normalization (e.g., to MNI space) [38,39,71,72,73,99,100,101,102,103,104,105,106,107], and (v) spatial smoothing using a Gaussian kernel with 2 mm [38], 4 mm [104], 6 mm [105,107], 7 mm [99], or 8 mm at full width at half maximum (FWHM) [39,71,72,73,100,101,102,106]. In addition to the mentioned “basic” processing steps, in several of the reviewed studies, more sophisticated filter methods were applied in order to account for artifacts. In four studies, high-pass filters using cut-off frequencies of 1/128 Hz [99,100,101,102] or 1/60 Hz [105] were applied, whereas in one study, a bandpass filter with a range of 0.008–0.08 Hz was used [107] to reduce the influence of specific sources of noise (e.g., high-frequency physiological artifacts or non-physiological low-frequency artifacts introduced by scanner drift). In two studies and one study, respectively, an AR(1) model [100,101] and an AR(2) model [102] were used to account for temporal autocorrelation. Furthermore, to correct motion-related artifacts, sophisticated motion correction methods, such as the MCFLIRT algorithm [105], slice-oriented motion correction algorithm (SLOMOCO) [103], or an independent-component analysis (ICA)-based approach for automatic removal of the motion artifacts (i.e., ICA-AROMA) [107] were used. A nuisance regression was conducted in several of the reviewed studies to regress out signals from the motion-related artefacts (e.g., [six-rigid] body [head] movements) [38,73,99,100,101,102] or physiological artifacts arising from the white matter [38,73,107] or cerebrospinal fluid [38,73,107]. In addition, three studies monitored physiological parameters, such as heart rate [38,72,107] and blood pressure, during fMRI scans [72].

The majority of the studies analyzed task-based brain activation patterns using the general linear model approach [38,39,71,72,99,100,101,102,104,105,106]. Furthermore, in the reviewed studies, functional connectivity was assessed during resting-state utilizing a seed-based approach [73] or a cross-correlation approach [107]; during a cognitive task, a psychophysiological interaction approach (PPI) was used [38]. 

### 3.5. Findings

(i) Main findings with regard to behavioral changes

In comparison to the control condition, significant exercise-induced improvements were observed (i) in executive functioning and working memory performance in healthy children after 30 min of moderate-intensity cycling [71,73], (ii) in a mnemonic discrimination task in healthy younger adults after 10 min of moderate-intensity cycling [38], (iii) in executive functioning in younger adults with ADHD after 30 min of moderate-intensity cycling [102], (iv) in working memory tasks in younger adults with low cardiorespiratory fitness level after 20 min of moderate-intensity cycling [106], (v) in executive functioning in healthy younger adults after circa 30 min of moderate-intensity and vigorous-intensity cycling [101], (vi) in a working memory task in healthy older adults after 20 min of light-intensity and moderate-intensity cycling [107], and (vii) in executive functioning in healthy older adults after 20 min of vigorous-intensity cycling [103]. 

In contrast, compared to the control conditions, non-significant changes in performance were observed (i) in an attentional task in adolescents with and without bipolar disorder after 20 min of moderate-intensity cycling [105], (ii) in an attentional task in healthy younger adults after 20 min of moderate-intensity cycling [72], (iii) in a working memory task in healthy younger adults after 20 min of moderate-intensity cycling [39], (iv) in a Go/No-Go task in younger adults with and without ADHD after 30 min of moderate-intensity cycling [100], (v) in a reward processing task in healthy younger adults after 30 min of moderate-to-vigorous treadmill running [99], and (vi) in a memory task in healthy older adults after 20 min of vigorous-intensity cycling [104]. A more detailed overview of the main findings regarding exercise-induced changes in behavioral performance is provided in Table 1.

(ii) Main findings with regard to changes in functional brain hemodynamics

While the evidence regarding behavioral changes is mixed, significant changes in functional brain hemodynamics in response to acute physical exercise are routinely observed (for a detailed overview, see Table 1). These exercise-induced functional hemodynamic brain changes were primarily found in the frontal (e.g., inferior and superior frontal gyrus) and temporal lobes (e.g., temporal gyrus, fusiform gyrus, and hippocampus). In particular, a higher activation (compared to the control condition) was observed (i) in healthy younger adults in response to 20 min of moderate-intensity cycling in the right middle frontal gyrus, the right lingual gyrus, and the left fusiform gyrus during a 2-back task [39]; (ii) in healthy younger adults in response to 30 min of moderate-intensity cycling in the left superior frontal gyrus, in the right precentral gyrus, and in the triangular part of the left inferior frontal gyrus during a Go/No-Go task [101]; (iii) in healthy younger adults in response to 30 min of high-intensity interval exercise in the left lingual gyrus and in the precuneus during a visual task [101]; (iv) in younger adults with ADHD in response to 30 min of moderate-intensity cycling in left and right middle occipital gyrus, right supramarginal gyrus, and left inferior parietal gyrus during correct inhibitions in a Go/No-Go task [100]; and (v) in healthy older adults in response to 20 min of moderate-to-vigorous-intensity cycling in the orbital part of the left inferior frontal gyrus, in the left inferior temporal gyrus, in the right middle temporal gyrus, in the left fusiform gyrus, and in the bilateral hippocampus during a semantic memory task [104], as well as in the orbital part of the left inferior gyrus and in the left inferior parietal gyrus during a Flanker task [103]. In contrast, a lower activation of the structures of the frontal and temporal lobe was observed (i) in adolescents with bipolar disorder after 20 min of moderate-intensity cycling in the orbital part of the left inferior frontal gyrus, the right frontal pole extending to temporal pole, the bilateral hippocampus, and the right amygdala during a Go/No-Go task [105] and (ii) in younger adults with ADHD after 30 min of moderate-intensity cycling in the right precentral gyrus, the right middle temporal gyrus, the left superior frontal gyrus, the right middle frontal gyrus, and the paracentral lobule during a Flanker task, and in the right superior frontal gyrus and the right middle frontal gyrus during a visual task [102]. However, especially in the cingulate gyrus, exercise-related deactivations were also observed in healthy adults. In particular, a significantly lower activation was observed in healthy younger adults after 20 min of moderate-intensity cycling in the anterior cingulate gyrus [39,106], the right paracentral lobule, and the left inferior frontal gyrus during a working memory task [39], and in the left parietal operculum during an attentional task [72]. Furthermore, in healthy younger adults, 30 min of moderate-intensity cycling led to a decrease of the activity of the left anterior cingulate gyrus during a visual task [101], whereas 30 min of moderate-to-vigorous treadmill running decreased the activation of the caudate nucleus during a reward processing task [99]. In healthy older adults, 20 min of moderate-to-vigorous-intensity cycling led to exercise-related deactivations in the right anterior cingulate gyrus during a Flanker task [103].

Furthermore, we found evidence in the reviewed studies that different exercise protocols (continuous cycling at moderate intensity versus high-intensity interval cycling) [101], as well as the cardiorespiratory fitness level [106] and sex of the participants [101], influence exercise-related changes in functional brain hemodynamics (see Table 1 for a detailed overview).

With regard to the effects of different exercise protocols, it was observed that the left superior frontal gyrus and right insula showed higher activation in the moderate-intensity group during a Go/No-Go task, whereas in the high-intensity group, the left lingual gyrus, the right precuneus, and the left anterior cingulate cortex exhibited higher activation during a visual task [101].

Regarding the effects of cardiorespiratory fitness level (CRF), fitness level-dependent changes mainly manifested in the different activation patterns of the cerebellum. In particular, in the group with a high level of CRF, the activation of the right cerebellum was higher before cycling than after cycling (0-back and 1-back condition), whereas the activation of the left anterior cingulate gyrus and the left globus pallidus (a 2-back condition) was higher after cycling compared to before cycling [106]. In the group with a low level of CRF, the activation of the right cerebellum was higher after cycling compared to before cycling (0-back and 1-back condition), while the activation of the left anterior cingulate gyrus was lower after cycling compared to before cycling (2-back condition) [106]. The group comparison of the high-fit individuals and the low-fit individuals revealed that in the younger adults with a high level of CRF, the right cerebellum showed higher activation before cycling (0-back and 1-back condition), while the left anterior cingulate gyrus and the left globus pallidus exhibited lower activation before cycling (2-back condition) [106]. In comparison to the younger adults with a low level of CRF, the high-fit younger adults exhibited a lower activation of the right cerebellum (0-back and 1-back condition) and higher activations of the left anterior cingulate gyrus and the left globus pallidum (2-back condition) after cycling [106].

With respect to sex-specific differences in the functional brain activation patterns, male participants in the moderate-intensity group showed higher activation of the right precentral gyrus during a Go/No-Go task compared to female participants [101].

In addition, in children, an exercise-related increase after 30 min of moderate-intensity cycling was observed in the left superior/inferior parietal gyrus, the right superior parietal gyrus, the left hippocampus, and the bilateral cerebellum during a working memory task [71]. 

Suwabe et al. [38], who utilized the PPI approach to investigate the effects of 10 min of moderate-intensity cycling on task-based connectivity patterns, observed an increase in the functional connectivity between the hippocampal subfields dentate gyrus (DG)/CA3 and the cortical areas located in the temporal lobe (i.e., parahippocampal cortex and fusiform gyrus) and the parietal lobe (i.e., angular gyrus) [38]. In particular, positive associations between the hippocampus and the left angular gyrus, the left fusiform gyrus, the left parahippocampal cortex, and the left primary visual cortex were observed, while negative associations between the hippocampus and the left temporal pole were noted [38].

(iii) Neurobiobehavioral associations

In healthy children, it was observed that after 30 min of moderate-intensity cycling, the increment of the resting-state functional connectivity between the left cerebellum and the right inferior frontal gyrus was negatively correlated with better cognitive performance [73].

Furthermore, in a cohort of healthy younger adults, Mehren et al. [101] observed positive correlations between cardiorespiratory fitness levels and activation of the right insula (during Flanker task) and the left rolandic operculum (during a Go/No-Go task) in the moderate-intensity group. In the same study, a negative correlation between cardiorespiratory fitness level and the activation of the right postcentral gyrus (during a Go/No-Go task) in the high-intensity interval group was also observed [101]. In addition, in healthy younger adults, it was noticed that higher correlations between the hippocampal subfield DG/CA3 and the angular gyrus, the fusiform gyrus, and the parahippocampal cortex predicted higher performance in the mnemonic discrimination task [38].

In older adults, Voss et al. [107] observed that acute changes in behavioral performance and resting-state functional connectivity in the hippocampal-cortical connections in response to 20 min of moderate-intensity cycling can predict long-term training-related neurocognitive changes (i.e., neurocognitive changes after 12 weeks of aerobic training). For instance, the authors provide evidence that exercise-related changes of the right-lateralized fronto–parietal connections are related to acute and long-term gains in working memory performance [107]. A more detailed overview of the main findings is provided in Table 1.

## 4. Discussion

### 4.1. Risk of Bias 

Overall, in most domains, the included studies were judged to have an unclear risk of bias (see Figure 3). In particular, in the domains of sequence generation, allocation concealment, blinding of participants and personnel, and blinding of outcome assessment, all included studies were rated as having an unclear risk of bias because they did not describe the used procedures in sufficient detail. Hence, we recommend that future studies should follow established guidelines (e.g., the CONSORT statement [112,113]) and report their study procedures in sufficient detail. Notably, the blinding of participants in exercise studies is exceedingly difficult because the participants notice if they are exercising or not. To minimize the risk of bias arising from not blinding participants (e.g., due to expectation effects), it is recommended that future studies should control appropriately for confounders, for instance, by assessing additional biopsychosocial variables (e.g., sleep, level of arousal, mood) [114,115,116,117,118] and/or a placebo group [116] (e.g., the placebo [sham] group performs the same exercise but without loading [119,120] or an inadequate dose to induce considerable effects [121]) when possible. Furthermore, with respect to the domain of “other bias”, we recommend that upcoming studies pay stronger attention to developing a more rigorous study design (see the next Section 4.2, Study characteristics, for a detailed discussion, as well as Figure 4) [33] and better exercise prescription (see Section 4.3, Exercise characteristics, for a detailed discussion). 

### 4.2. Study Characteristics

An appropriate study design is crucial when the effects of acute physical exercise on cognition and their underlying neurobiological correlates are to be investigated (e.g., changes in functional brain activation). In the majority of the reviewed studies, a within-subject design with crossover posttest comparisons was used to contrast the brain activation patterns obtained after rest and after an acute bout of physical exercise. Studies using this type of within-subject design with crossover posttest comparisons, however, fail to account for day-to-day variations in factors such as sleep quality, which is known to influence cognitive performance [122,123,124,125] and brain activation [126,127,128]. Hence, using a within-subject design with crossover posttest comparisons makes it difficult to determine to what extent the observed changes were induced by the physical intervention itself or were a consequence of day-to-day variations [33]. To overcome these drawbacks, future investigations should make use of a within-subject crossover design with a pretest-posttest comparison (see Figure 4). This approach minimizes the influence of confounding factors and strengthens the validity of the observed effects [33].

Another crucial factor is the period between the cessation of the physical exercise and the administration of the cognitive tests (see Table 1). Behavioral results make it clear that, among other factors (e.g., exercise intensity), the length of this rest period affects cognitive performance substantially [30]. The neurobiological processes underlying this break effect (e.g., functional brain activation), however, have only been sparsely investigated [30,33]. Thus, studies investigating how brain activation is affected by the duration of the rest period (e.g., 10 min versus 60 min or 10 min versus the time needed to reach a level within 10% of the participant’s resting heart rate) are necessary to deepen our understanding of exercise–cognition interaction. Notably, fMRI scans require that every subject be prepared properly. This impedes immediate scans after the cessation of physical exercise. Fortunately, such delays between physical exercise and cognitive testing increase the effect sizes for cognitive improvements [30]

The experimental design of fMRI studies can be classified according to the following categories: (i) resting-state (e.g., analyzing fMRI data obtained while no explicit task is performed) or (ii) task-based (e.g., analyzing fMRI data obtained while a specific task is performed) [62]. While a resting-state fMRI commonly involves the assessment of functional connectivity (e.g., the temporal correlation between time series of different brain regions), in a task-based fMRI, either the magnitude of the cerebral hemodynamic response (e.g., using GLM approach) or functional connectivity (e.g., using PPI approach) are measured [62]. The majority of the reviewed studies conducted a task-based fMRI and used either a block design [39,71,72,101,102,105,106] or an event-related design [38,99,101,102,103,104,107]. Each design offers advantages and disadvantages [74,129]. The block design is a powerful paradigm to detect neuronal activity but (i) is limited with regard to its evaluation of temporal characteristics [129], (ii) can be influenced by canceling effects [130], and (iii) does not offer the possibility to perform a trial-to-trial analysis [74,129]. An event-related design provides valuable information about the temporal characteristics of neuronal activity, but many trials are needed for averaging to achieve an adequate signal-to-noise ratio [131] and appropriate statistical power [129,132]. A mixed design (consisting of a mixture of block design and event-related design) allows the investigation of ‘maintained’ versus ‘transient’ neural activity, but is challenging in terms of estimating the hemodynamic response function [74,129,133,134,135]. However, an in-depth discussion of all the advantages and disadvantages of different cognitive tests, fMRI experimental designs, and fMRI analysis methods is beyond the scope of this review; the reader may find valuable related information in the literature [62,74,129,133]. 

In summary, the advantages and disadvantages of the study design (e.g., within-subject crossover design with pretest-posttest comparisons), experimental fMRI design (e.g., block design), and fMRI analysis methods (e.g., PPI approach) should be carefully considered to address the specific research aim(s) appropriately.

### 4.3. Exercise Characteristics

The prescription of an acute bout of physical exercise is characterized by the following exercise variables: (i) exercise intensity, (ii) exercise duration and (iii) type of exercise (e.g., dancing or cycling) [33,136]. In this regard, it is important to acknowledge that exercise intensity can be specified and operationalized by using either parameters of external load (e.g., cycling while sustaining a distinct level of mean power output, i.e., in watts) or markers of internal load (e.g., cycling at a distinct percentage of the subject’s maximum heart rate) [137,138,139,140,141,142,143,144,145]. While the external load is characterized by the work completed by an individual independent of internal characteristics (e.g., the mean power output in watts over a distinct time duration and at a specific cadence), the internal load (e.g., heart rate, ratings of perceived exertion, or blood lactate response) is characterized by the psychophysiological response(s) to the external load and the influencing factors, such as personal characteristics (e.g., training status, sleep, nutrition, or genetics) and environmental factors (e.g., room temperature) [137,138,139,140,141,142,143,144,145]. In the literature, the product of exercise intensity, exercise duration, and type of physical exercise is commonly referred to as the dose [33,136], whereas a newer approach refines this definition and proposes that this dose can be objectified by using a specific marker(s) of the internal load as a proxy [29,138]. However, the dose–response relationship between acute physical exercise, cognition, and the underlying neurobiological processes (e.g., changes in cerebral hemodynamics) is currently poorly understood [26,33,35,146]. While one study provides clear evidence that brain activation patterns and cognitive performance are influenced by the exercise protocol [101], the remaining reviewed studies are too heterogeneous with regard to their studied exercise variables (e.g., exercise intensity and exercise duration) and too homogeneous with regard to the type of physical exercise they examine (only endurance exercises, such as stationary cycling or treadmill running) to derive solid conclusions in this direction. Cognizant of these limitations, further studies utilizing fMRI are required to investigate the effects of different exercise variables (e.g., different exercise intensities) or types of exercise (e.g., resistance exercise) on cognitive performance and the underlying functional brain activation changes. The investigation of the effects of acute resistance exercises [147,148] and/or acute coordinative/complex/motor-cognitive exercises [149,150,151] on cerebral hemodynamics (e.g., measured by fMRI) during a standardized cognitive test seem especially promising since these types of physical exercise have been demonstrated to lead to behavioral improvements in tests of attention, executive functions, and working memory. In addition, future studies should pay more attention to a more accurate prescription of exercise intensity because eight out of fourteen studies reviewed determined exercise intensity by using specific formulas to calculate the target heart rate [39,71,72,73,105,106] or used subjective ratings of the perceived exertion [102,103]. In particular, the use of specific formulas is less than optimal because such an approach cannot accurately predict exercise intensity [152,153,154,155,156,157,158]. Thus, in order to determine exercise intensity (or load) more accurately, it is recommended to conduct a graded exercise test for endurance exercise or to quantify the one-repetition maximum (1-RM) for resistance exercises in a separate testing session. Furthermore, it is recommended to pay stronger attention to an adequate exercise prescription, and we advocate for the reporting of both parameters of external load and markers of internal load in order to characterize the exercise protocol adequately [29,137,138,159]. By doing so, the comparability and reproducibility of the research results are very likely to be increased [137,138].

### 4.4. Data Processing

Measuring “true” brain activation is challenging because the BOLD signal can be influenced by non-neuronal confounders that arise from (i) instrumental noise, (ii) motion-related artifacts, and (iii) physiological artifacts [160,161,162,163,164,165,166,167]. Hence, in order to minimize the influence of artifacts and to achieve an optimal signal-to-noise and contrast-to-noise ratio, an appropriate (pre-)processing of the fMRI data is necessary. While there is currently no “gold standard” available for the processing of fMRI data [62,168], based on the analysis of the data processing procedures in the studies reviewed, it is recommended to incorporate the following “basic” data processing steps: (i) discard the first three to twelve images to avoid magnetization disequilibrium and allow the participants to habituate themselves to the scanner noise; (ii) employ slice-time correction to account for the time difference of the acquisition of each slice; (iii) utilize realignment to correct for subject motion; (iv) use intra-subject inter-modal registration (e.g., to a T1-weighted image) and spatial normalization (e.g., to MNI space); (v) apply spatial smoothing, which helps, for instance, to cope with interindividual anatomical variability and improve the signal-to-noise ratio; and (vi) apply denoising to remove the confounders and artifacts at known frequencies (e.g., scanner drifts ~< 0.01 Hz; cardiac artifacts ~0.15 Hz; respiratory artifacts ~0.3 Hz [62]). With regard to temporal filtering, only five out of the fourteen reviewed studies reported their detailed cut-off frequencies used for temporal filtering [99,101,102,105,107]. Thus, in order to enhance reproducibility, further studies should pay more attention to a detailed description of methodological details (e.g., used cut-off frequencies), thereby taking the published guidelines into account [75]. In addition to the mentioned “basic” processing steps, the correction of motion artifacts and physiological artifacts is necessary to achieve a relatively high signal quality. Artifacts arising from motion are serious confounders in fMRI studies and, thus, should be minimized [62,161,169,170,171]. The reviewed studies accounted for motion by using motion parameters as regressors or by using data-driven approaches (e.g., ICA-based methods). Indeed, the application of motion-correction techniques provides considerable improvements in data quality [169,172,173,174,175]. There seems to be no significant quality difference between the motion correction tools integrated into different popular software solutions [175,176]. However, it should be emphasized that the best approach to deal with motion artifacts is currently to reduce them as much as possible by, for instance, a comfortable padding and head fixation [62,177,178].

With regard to acute physical exercise, physiological artifacts that stem from changes in heart rate [179,180,181] and/or breathing [180,182,183,184,185,186,187,188,189] are a major challenge because acute physical exercise induces strong changes in systemic physiology (e.g., an increase in heart rate), which, in turn, may confound the BOLD signal substantially. To minimize the effects of such physiological artifacts on the BOLD signal, sophisticated cleanup techniques are needed. The available cleanup techniques can be classified as data-based approaches and approaches that rely on recordings of additional physiological parameters (e.g., cardiac or respiratory parameters) [160,164]. Since these cleanup procedures (e.g., using additional physiological measures, such as cardiac or respiratory signals as regressors) hold great potential to improve data quality [181,184], future studies should consider and incorporate them by default. It is strongly recommended to check the MR compatibility of the equipment used to quantify additional physiological parameters in order to avoid side effects [190,191]. More detailed information on how to account for different types of artifacts and noise can be found in the referenced literature [62,164,169,170,175,192,193,194,195,196,197,198]. All reviewed studies used software packages such as “Statistical parametric mapping” (SPM) or “Analysis of Functional NeuroImages” (AFNI) to analyze their fMRI data (see Table 1). A comprehensive overview of the main features of these common fMRI software packages is provided by Soares et al. [62]. 

### 4.5. Findings

There is growing evidence that acute bouts of aerobic exercise [30,199] or resistance exercise [147,148] improve cognitive performance. The fMRI studies reviewed here only partially support this notion because they report mixed findings (both improvements in behavioral performance [38,71,73,101,102,106,107] and non-significant changes in behavioral performance [39,72,99,100,104,105] were observed). Such heterogeneity may arise from the great variability in the applied research protocols, wherein it is known that behavioral improvements in response to acute physical exercise are dependent on different factors, such as the characteristics of participants (e.g., age), the cognitive domains tested (e.g., executive functions), the exercise characteristics (e.g., exercise intensity), and the time delays between the cessation of exercise and post-exercise cognitive testing [30,147,148,199]. 

However, independent of the degree of behavioral improvements, almost all reviewed studies reported profound changes in brain activation (especially in the frontal lobe, the cerebellum, and the hippocampus) in response to exercise (see Table 1). In general, the observations of a higher activation in distinct regions of the frontal lobe and cognitive improvements after exercise cessation are in line with the findings provided by other functional neuroimaging techniques (i.e., fNIRS) [34]. 

Furthermore, the results of Li et al. [106] and Mehren et al. [101] suggest that changes in functional brain activation patterns depend on (i) the cardiorespiratory fitness level, (ii) the exercise protocol, and (iii) sex. Although the number of fMRI studies in this area is limited, the moderating roles of (i) cardiorespiratory fitness levels [30], (ii) exercise intensity [30,200], and (iii) sex [30] are in accordance with the literature in summarizing the effects of acute physical exercise on behavioral performance. Thus, there is good reason to assume that investigations using fMRI can help to elucidate the effects of the mentioned moderators on neural correlates. The crucial role of using neuroimaging techniques (e.g., fMRI) to understand the neural processes involved in the interaction of acute exercise with cognition is underlined by the observed neurobehavioral associations. For instance, in three studies, correlations between changes in brain activation and behavioral performance were noticed [38,73,107], and in one study, correlations between cardiorespiratory fitness level and brain activation were observed [101]. However, based on the small number of studies using fMRI and the great variability between research protocols, it is too early to draw generalizable conclusions. Cognizant of these limitations, more comprehensive and well-designed trials are needed to utilize the possibilities provided by fMRI to understand exercise–cognition interaction. 

Such upcoming studies could evaluate whether exercise-induced changes in neurotransmitter release and/or neurometabolic changes (e.g., by using positron emission tomography [PET]) [201,202,203,204] are related to exercise-induced improvements in cognitive performance. In this regard, PET studies suggest that during high-intensity exercise, lactate, which is released by the working muscles during physical exercise and can cross the blood–brain barrier via monocarboxylate transporters [205,206,207,208,209,210,211,212], is increasingly utilized by the brain to satisfy the increased energy demand of neural tissue [213]. Interestingly, after acute high-intensity interval exercises, changes in blood lactate levels have been linked to performance changes in executive functioning [214,215]. Based on these findings, further PET studies focusing on the associations between exercise-related changes in blood lactate and cognitive performance could provide neuroimaging evidence for the underlying neurobiological processes of acute exercise-related cognitive improvements. Furthermore, future studies could also assess whether acute physical exercise modulates hemodynamic changes in white matter during a cognitive task because there is growing evidence that acute hemodynamic changes occur in the white matter while solving cognitive tasks [216,217,218,219,220,221,222,223]. Another parameter worthy of examination in acute exercise-cognition studies is the (within-person) brain signal variability, which provides new insights into cognition-related processes of the human brain [224,225,226,227,228,229,230,231,232,233]. Furthermore, to derive valid and reliable conclusions in repeated measurements, a high level of reproducibility is necessary [234,235,236]. With regard to the assessment of cognitive performance prior to and after an acute bout of physical exercise, a considerable amount of test–retest errors for specific measures of cognition have been reported [237], suggesting that participants should be appropriately habituated to the whole testing procedure. Concerning the reproducibility of fMRI, it was observed that reproducibility is affected by experimental factors, such as the length of the scan [238] and the design of the cognitive task [239]. However, currently, there is no study available that investigates the reproducibility of exercise-induced functional brain activation changes (e.g., measured by fMRI). Such an evaluation of reproducibility is needed to improve our interpretations of the obtained results. In addition, there is some evidence that several biopsychosocial factors (e.g., circadian rhythms [240,241,242], level of sleepiness [243], and level of arousal [244]) can influence cognitive performance. In particular, it was observed (i) that measures of the cortical hemodynamic response (e.g., obtained during a cognitive test) are associated with total sleep duration [245,246] and sleepiness [247,248] and (ii) that total sleep duration is associated with acute exercise-related changes in behavioral performance (e.g., executive function performance) [82]. Interestingly, the relationship between total sleep duration and acute exercise-related changes in executive function performance are mediated by the volume of the caudate nuclei [82]. However, based on the evidence that cognitive performance is influenced by several biopsychosocial factors (e.g., sleep and arousal), future studies should consider these factors in order to elucidate their effects on acute exercise-related changes in cognitive performance and functional brain activation.

## 5. Conclusions

In summary, this review provides an overview of fMRI applications in the investigation of the effects of acute physical exercise on cognitive performance and their underlying neural correlates (e.g., changes in functional brain activation patterns). Although the small number of studies available and the great variability across study protocols limits the number of conclusions that can be drawn with certainty, our observations suggest that acute physical exercise (e.g., cycling) leads to profound changes in functional brain activation, while the evidence with regard to behavioral changes is equivocal. Thus, further well-designed trials are needed to strengthen our understanding of acute exercise-induced changes in functional brain activation, including an investigation of a possible dose–response relationship. Future studies should consider (i) a more rigorous study design (e.g., within-subject crossover design with a pretest-posttest comparison), (ii) the integration of sophisticated filter methods in fMRI data analysis (e.g., to account for physiological artifacts), (iii) a more detailed description of the applied processing steps in fMRI data analysis (e.g., the applied filter cut-off frequencies and the atlas used for neuroanatomical labeling), and (iv) a more precise exercise prescription (e.g., including the reporting of both parameters of external load and markers of internal load).

## Figures and Tables

**Figure 1 brainsci-10-00175-f001:**
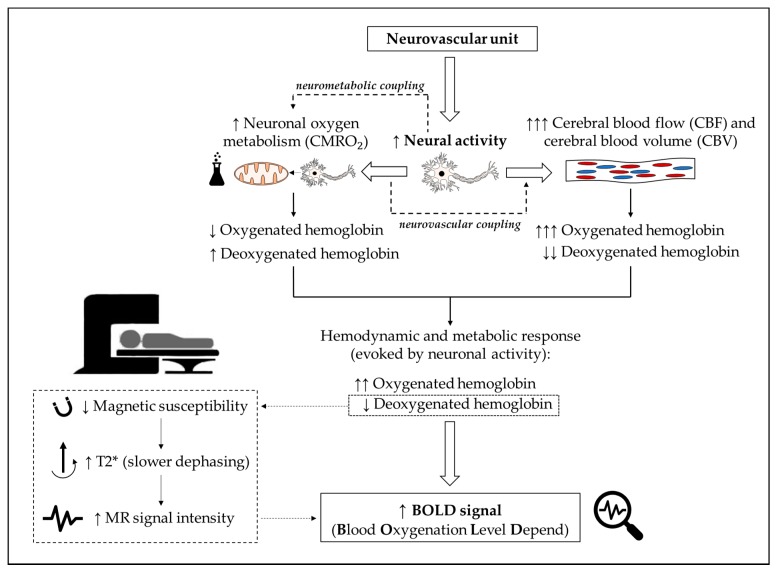
Schematic illustration of the neurovascular unit and changes in cerebral hemodynamics and oxygenation induced by neural activity (e.g., as a consequence of solving a cognitive task). Furthermore, in the lower left part of the illustration, the basic physical principles of magnetic resonance imaging using the BOLD contrast (Blood Oxygenation Level Depend) are displayed.

**Figure 2 brainsci-10-00175-f002:**
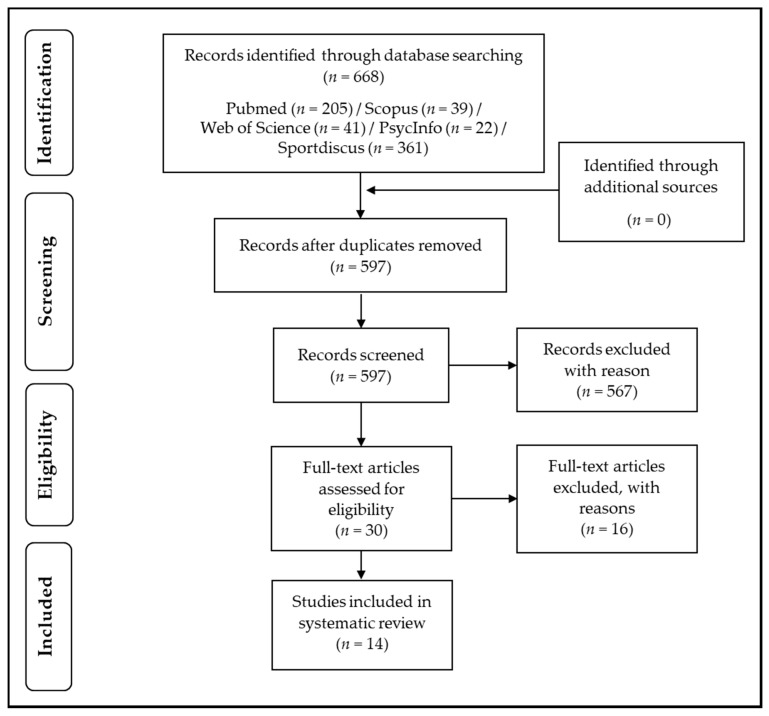
Flow chart with information about the search, screening, and selection processes that led to the identification of the relevant articles for inclusion in this systematic review.

**Figure 3 brainsci-10-00175-f003:**
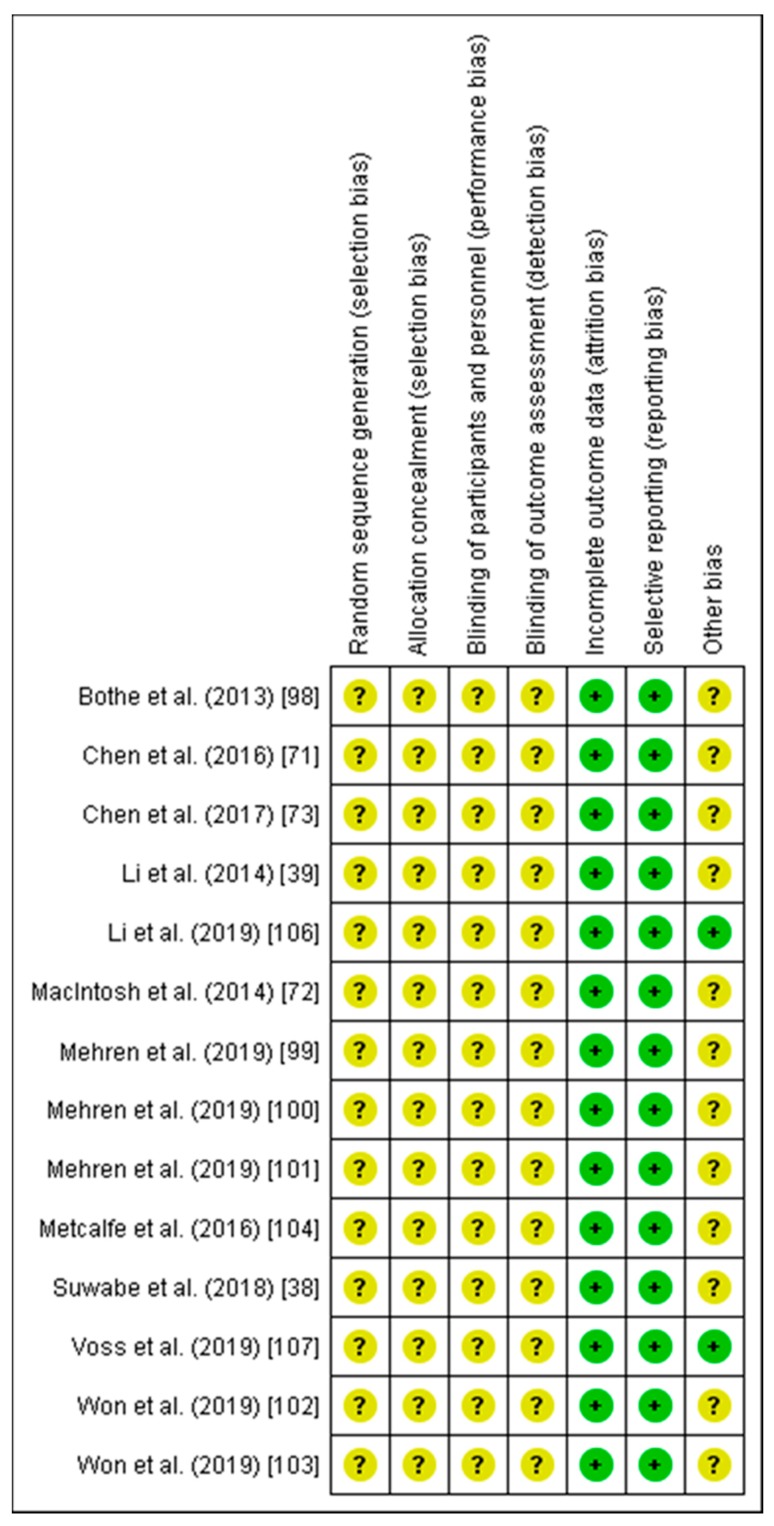
Illustration of the analysis of the risk of bias assessment of the included studies. This figure was created by using Review Manager [98] and follows the recommendations outlined in the Cochrane Collaboration guidelines [97]. A “green plus” indicates a low risk of bias and a “yellow question mark” indicates an unclear risk of bias.

**Figure 4 brainsci-10-00175-f004:**
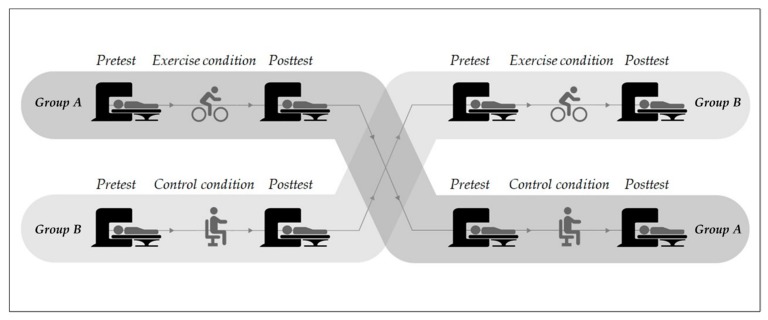
Schematic illustration of the recommended study design (a within-subject crossover design with a pretest-posttest comparison).

**Table 1 brainsci-10-00175-t001:** Overview of the selected population characteristics and main findings of the reviewed studies.

First Author (year)	Sample Characteristics and Study Design (1.) Health Status (2.) Cardiorespiratory Fitness Level (mean ± SD in ml/kg/min) (3.) Number of Participants (gender) (4.) Age (mean ± SD or range in years) (5.) (i) Body Height (mean ± SD or range in cm)/ (ii) Body Mass (mean ± SD or range in kg)/ (iii) BMI (mean ± SD)	Main Findings (1.) Behavioral Level (2.) Functional Level
Bothe et al. (2013) [99]	(1.) Young healthy trained and inactive adults (2.) HT (T) 56.0 ± 5.2; HT (P) 55.1 ± 6.2/ PIA (T) 48.4 ± 6.8; PIA (P) 50.2 ± 2.9 (3.) HT (T) N = 9 (9 m); HT (P) N = 10 (10 m)/ PIA (T) N = 11; PIA (P) N = 13 (13 m) (4.) HT (T) 24.0 ± 4.0; HT (P) 26.1 ± 3.4/ PIA (T) 25.1 ± 3.0; PIA (P) 23.9 ± 2.7 (5.) (i) N.A./ (ii) N.A./ (iii) HT (T) 22.8 ± 2.0; HT (P) 23.0 ± 1.9, PIA (T) 22.4 ± 1.5; PIA (P) 21.6 ± 1.5	*Treadmill running vs. placebo (stretching):* (1.) - no significant differences (2.) ↓ lt. caudate nucleus in gain anticipation ↓ rt. caudate nucleus in gain feedback
Chen et al. (2016) [71]	(1.) Healthy children (2.) N.A. (3.) N = 9 (4 f/ 5 m) (4.) 10.0 ± 0.0 (5.) (i) N.A./ (ii) N.A./ (iii) N.A.	*Cycling vs. seated rest:* (1.) ↓ mean RT in the 2-back condition (2.) ↑ lt. superior/inferior parietal gyrus, rt. superior parietal gyrus, lt. hippocampus, lt. cerebellum, rt. cerebellum
Chen et al. (2017) [73]	(1.) Healthy children (2.) N.A. (3.) N = 9 (4 f/ 5 m) (4.) 10.0 ± 0.0 (5.) (i) N.A./ (ii) N.A./ (iii) N.A.	*Cycling vs. seated rest:* (1.) ↓ mean RT in Eriksen flanker task (2.) ↑ resting-state functional connectivity in the lt. cerebellum-rt. inferior gyrus is significantly negatively correlated with better performance (e.g., shorter reaction time) in an Eriksen flanker task
Li et al. (2014) [39]	(1.) Healthy younger adults (2.) N.A. (3.) N = 15 (15 f) (4.) 19-22 (5.) (i) 161.6 ± 3.1/ (ii) 50.8 ± 3.4/ (iii) 19.5 ± 1.4	*Cycling vs. seated rest:* (1.) - no significant differences (2.) ↑ rt. middle frontal gyrus, rt. lingual gyrus, lt. fusiform gyrus/↓ anterior cingulate gyrus, rt. paracentral lobule, and lt. inferior frontal gyrus in the 2-back condition
Li et al. (2019) [106]	(1.) Healthy high-fit and low-fit younger adults (2.) HF 26.5 ± 1.9/ LF 19.9 ± 1.0 (3.) HF N = 12 (12 f)/ LF N = 12 (12 f) (4.) HF 25.5 ± 0.7/ LF 25.8 ± 0.6 (5.) (i) HF 164.8 ± 5.9; LF 161.1 ± 4.9/ (ii) HF 55.7 ± 5.1; LF 52.0 ± 5.0/ (iii) HF 20.5 ± 1.1; LF 20.0 ± 1.5	*After cycling vs. prior cycling:* (1.) ↑ accuracy in LF in 0-back ↓ accuracy in HF in 1-back (2.) ↓ rt. cerebellum in HF in 0-back and 1-back ↑ rt. cerebellum in LF in 0-back and 1-back ↑ lt. and rt. cerebellum, rt. paracentral lobule, and rt. medial temporal pole in HF and LF in 1-back ↑ lt. anterior cingulate gyrus and in lt. pallidum in HF in 2-back ↓ lt. anterior cingulate cortex in LF in 2-back *HF vs. LF:* (1.) ↑ accuracy in 1-back and 2-back (before cycling) (2.) ↑ rt. cerebellum in 0-back (before cycling) ↓ rt. cerebellum in 0-back and 1-back (after cycling) ↑ lt. inferior parietal lobule in 0-back and 1-back ↓ lt. anterior cingulate gyrus and in lt. globus pallidus in 2-back (before cycling) ↑ lt. anterior cingulate gyrus and in lt. globus pallidus in 2-back (after cycling) ↑ rt. superior frontal gyrus in 2-back
MacIntosh et al. (2014) [72]	(1.) Healthy younger adults (2.) N.A. (3.) N = 16 (10 f/ 6 m) (4.) 26.7 ± 4.1 (20-35) (5.) (i) 171.0 ± 10.5/ (ii) 67.9 ± 13.9/ (iii) N.A.	*After vs. prior cycling:* (1.) - no significant change (2.) ↓ lt. parietal operculum
Mehren et al. (2019) [100]	(1.) Younger adults with and without ADHD (2.) ADHD 36.6 ± 7.5/ HC 42.0 ± 7.3 (3.) ADHD N = 23 (3 f/ 20 m)/ HC N = 23 (4 f/ 20 m) (4.) ADHD 31.4 ± 9.6/HC 29.5 ± 7.0 (5.) (i) N.A./ (ii) N.A./ (iii) ADHD 25.6 ± 4.3; HC 24.1 ± 2.6	*Cycling vs. rest (watching movie):* (1.) – no significant differences (2.) ↑ lt. middle occipital gyrus, rt. middle occipital gyrus, rt. supramarginal gyrus, and lt. inferior parietal gyrus during correct inhibitions in Go/No-Go task in ADHD - negative correlation between exercise-related changes in brain activation in lt. insula, lt. precentral gyrus, and rt. postcentral gyrus during correct inhibitions in Go/No-Go task and task performance in the control condition in ADHD *ADHD vs. HC* (1.) – no significant differences (2.) ↑ lt. superior occipital gyrus, rt. precuneus, and lt. supramarginal gyrus
Mehren et al. (2019) [101]	(1.) Younger healthy adults (2.) MIE 39.6 ± 7.1/ HIE 37.0 ± 8.3 (3.) MIE N = 32 (16 f/ 16 m)/ HIE N = 31 (16 f/ 15 m) (4.) MIE 29.3 ± 8.5/ HIE 28.6 ± 7.7 (5.) (i) N.A./ (ii) N.A./ (iii) MIE 23.8 ± 2.3; HIE 24.5 ± 4.8	*Cycling vs. rest (watching movie):* (1.) ↓ reaction times in a congruent and incongruent condition (Flanker task) in HIIE ↑ sensitivity index (Go/No-Go task) in MICE (2.) ↑ lt. superior frontal gyrus, rt. precentral gyrus and the triangular part of the lt. inferior frontal gyrus in Go/No-Go task (contrast “hits”) in MICE ↓ lt. anterior cingulate gyrus in a visual task in MICE ↑ lt. lingual gyrus and precuneus in a visual task in HIIE ↑ rt. precentral gyrus in Go/No-Go task (contrast “hits”) in MICE (males vs. females) *MICE vs. HIIE:* (1.) - no significant differences (2.) ↑ lt. superior frontal gyrus and rt. insula in Go/No-Go task (contrast “hits”) ↓ lt. lingual gyrus, rt. precuneus, and lt. anterior cingulate gyrus in a visual task *Associations between brain activation and CRF:* - positive association between activation of rt. insula and VO_2 peak_ in MICE (Flanker task contrast “incongruent – congruent”) - negative association between activation of rt. postcentral gyrus VO_2 peak_ in HIIE (Go/No-Go task – contrast “hits”) - positive association between activation of lt. rolandic operculum VO_2 peak_ in MICE (Go/No-Go task – contrast “correct inhibitions - hits”)
Mehren et al. (2019) [102]	(1.) Younger adults with and without ADHD (2.) ADHD 37.1 ± 7.2/ HC 41.5 ± 7.3 (3.) ADHD N = 20 (4 f/ 16 m)/ HC N = 20 (5 f/ 15 m) (4.) ADHD 29.9 ± 9.5/ HC 29.0 ± 7.4 (5.) (i) N.A./ (ii) N.A./ (iii) ADHD 25.0 ± 3.8; HC 24.3 ± 2.7	*Cycling vs. rest (watching movie):* (1.) ↓ interference score (incongruent-congruent) in HC ↓ mean RT in a congruent and incongruent condition in ADHD ↓ RT variability in a congruent condition in ADHD - significant positive correlation between RT differences (i.e., differences between exercise and control condition) in incongruent trials and VO_2 peak_ in ADHD (2.) ↓ rt. precentral gyrus and rt. middle temporal gyrus in a congruent condition in ADHD ↓ rt. superior frontal gyrus, rt. middle frontal gyrus and paracentral lobule in incongruent condition in ADHD ↓ rt. superior frontal gyrus and rt. middle frontal gyrus in a visual task in ADHD
Metcalfe et al. (2016) [105]	(1.) Adolescents with and without BD (2.) N.A. (3.) BD N = 30 (17 f/13 m)/ HC N = 20 (11 f/9 m) (4.) BD 16.8 ± 1.4/ HC 16.1 ± 1.4 (5.) (i) N.A./ (ii) N.A./ (iii) N.A.	*BD vs. HC:* (1.) – no significant differences (2.) ↓ orbital part of the lt. inferior frontal gyrus, rt. frontal pole extending to temporal pole, rt. and lt. hippocampus and rt. amygdala in Go trials
Suwabe et al. (2018) [38]	(1.) Healthy younger adults (2.) 37.9 ± 8.2 (3.) N = 16 (12 f/4 m) (4.) 21.1 ± 2.0 (5.) (i) 164.2 ± 9.1/ (ii) 55.4 ± 7.7/ (iii) 20.5 ± 1.6	*Cycling vs. seated rest:* (1.) ↑ better performance in mnemonic discrimination task (in high- and medium-similarity lures) (2.) - significant positive associations between dentate gyrus/CA3 subfield bilaterally and lt. angular gyrus, lt. fusiform gyrus, lt. parahippocampal cortex, and lt. primary visual cortex - significant negative association between dentate gyrus/CA3 subfield bilaterally and lt. temporal pole - significant association between the whole hippocampus (CA1, subiculum and dentate gyrus/CA3) and bilateral parahippocampal cortex - higher correlations between dentate gyrus/CA3 and angular gyrus, fusiform gyrus as well as parahippocampal cortex predicted improvements in cognitive performance
Voss et al. (2019) [107]	(1.) Healthy older adults (2.) 20.1 ± 5.0 (3.) N = 34 (20 f/ 14 m) (4.) 67.1 ± 4.3 (60-80) (5.) (i) N.A./ (ii) N.A./ (iii) 29.1 ± 5.3	*After vs. prior cycling:* (1.) ↑ accuracy in 1-back and 2-back condition in LIE and MIE and ↑ accuracy in 1-back and 2-back condition in MIE vs. LIE - acute improvements in working memory performance predict long-term improvements in working memory performance (after 12-weeks of training) (2.) - acute changes in the following connections predict long-term changes in the same connections: (i) lt. middle occipital lobule and rt. angular gyrus, (ii) lt. superior frontal gyrus and lt. inferior temporal gyrus, (iii) lt. superior frontal gyrus and rt. inferior parietal gyrus, (iv) rt inferior parietal gyrus and rt. triangular part of the inferior frontal gyrus, (v) rt. inferior parietal gyrus and opercular part of the lt. inferior frontal gyrus, (vi) lt. lingual gyrus and rt. fusiform gyrus, (vii) rt. superior frontal gyrus and rt. middle temporal gyrus, (viii) rt. middle temporal gyrus and rt. inferior parietal gyrus, (ix) lt. and rt. precuneus, (x) lt. superior frontal gyrus and lt. medial superior frontal gyrus, (xi) rt. middle temporal gyrus and opercular part of the lt. inferior frontal gyrus, and (xii) rt. superior frontal gyrus and rt. inferior parietal gyrus - acute and long-term changes in (i) connections between rt. middle temporal gyrus and rt. superior frontal gyrus, and (ii) connections between rt. inferior parietal gyrus and opercular part of the rt. inferior frontal gyrus are associated with performance in the 2-back condition
Won et al. (2019) [103]	(1.) Healthy older adults (2.) N.A. (3.) N = 32 (24 f/ 8 m) (4.) 66.2 ± 7.3 (55-85) (5.) (i) 166.6 ± 9.0/ (ii) 71.3 ± 14.2/ (iii) 25.5 ± 4.1	*Cycling vs. seated rest:* (1.) ↑ accuracy in a congruent and incongruent condition (2.) ↑ orbital part of the lt. inferior frontal gyrus and lt. inferior parietal gyrus in incongruent and incongruent-congruent condition ↓ rt. cingulate gyrus in incongruent and incongruent-congruent condition
Won et al. (2019) [104]	(1.) Healthy older adults (2.) N.A. (3.) N = 26 (20 f/ 6 m) (4.) 65.9 ± 7.2 (55-85) (5.) (i) 166.5 ± 8.5/ (ii) 73.6 ± 14.1/ (iii) 26.1 ± 4.3	*Cycling vs. seated rest:* (1.) - no significant differences (2.) ↑ orbital part of lt. inferior frontal gyrus, lt. inferior temporal gyrus, rt. middle temporal gyrus, lt. fusiform gyrus, and rt. and lt. hippocampus

ADHD: attention deficit hyperactivity disorder; BD: bipolar disorder; BMI: body mass index; cm: centimeter; f: female; HC: healthy controls; HF: high-fit group; HIIE: high-intensity interval exercise group; HT: highly trained individuals; kg: kilogram; LF: low-fit group; lt.: left; m: male; MICE: moderate-intensity continuous exercise group; min: minutes; N: number; N.A.: not applicable; P: placebo exercise (stretching and gymnastic tasks); PIA: physically inactive individuals; rt.: right; RT: reaction time; SD: standard deviation; T: treadmill exercise; VO_2 max_: highest value of maximal oxygen uptake attainable by a subject [109,110,111]; during cardiorespiratory fitness test; VO_2 peak_: highest “system-limited” oxygen uptake attained during the cardiorespiratory fitness test [109,110,111]; vs.: versus; ↑: significant increase; ↓: significant decrease.

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
