# Peer review of "The Contribution of Functional Magnetic Resonance Imaging to the Understanding of the Effects of Acute Physical Exercise on Cognition"

_brainsci, 2020, doi:10.3390/brainsci10030175_

Round 1

Reviewer 1 Report

The review of the contribution of fMRI to understand the effect of acute physical exercise on cognition is an interesting review article. Only a few minor commentaries.

In lines 50-51: "in the context of physically exercising is only...", maybe it is better to write: in the context fo physical exercise is only...

Line 316: "... Mehren et al [96] positive correlations between...", maybe it is better: ... Mehren et al [96] shown positive correlations...

Line 317: "...of the right insula (during Flanker task) or the left...", is "or" our "and"?.
Maybe it is better: ...of the right insula (during Flanker task) and the left...

Table 1: In the first column the year does not appear in any author. Please put the year, not only the numeric reference.

Page 12, Table 1: In the file of Mehren et al. [95], in the "Main findings" column, there is some typographical mistake: "-significant positive correlation between RT difference and ????????????...."

In the references, the journals of some articles are not presented with the abbreviation, maybe it can´t because there is not an abbreviation for these journals, but please, check it. I find them in lines: 618, 636, 644, 647, 660, 670, 678, 685, 691, 697, 741, 742, 747, 758, 885, 887, 903, 999, 1053, 1056, 1103. And maybe in others, I did not see.

Author Response

In lines 50-51: "in the context of physically exercising is only...", maybe it is better to write: in the context fo physical exercise is only...

  • We thank the reviewer for this valuable hint and edited the sentences as suggested.

“The most common methods used to investigate effects on functional brain activation are functional near-infrared spectroscopy (fNIRS) [34] and electroencephalography (EEG) [36,37]; however functional magnetic resonance imaging (fMRI) is only starting to be applied in the context of physical exercise [38,39].”

Line 316: "... Mehren et al [96] positive correlations between...", maybe it is better: ... Mehren et al [96] shown positive correlations...

  • We are thankful for this language advice and changed the sentence as suggested.

“Furthermore, in a cohort of healthy younger adults Mehren et al. [98] observed positive correlations between cardiorespiratory fitness levels and activation of the right insula (during Flanker task) and the left rolandic operculum (during a Go/No-Go task) in the moderate-intensity group.”

Line 317: "...of the right insula (during Flanker task) or the left...", is "or" our "and"?.

Maybe it is better: ...of the right insula (during Flanker task) and the left...

  • We changed the sentence as recommended by the reviewer.

“Furthermore, in a cohort of healthy younger adults Mehren et al. [98] observed positive correlations between cardiorespiratory fitness levels and activation of the right insula (during Flanker task) and the left rolandic operculum (during a Go/No-Go task) in the moderate-intensity group.”

Table 1: In the first column the year does not appear in any author. Please put the year, not only the numeric reference.

  • We thank the reviewer for this important hint and added the year in Table 1.

Page 12, Table 1: In the file of Mehren et al. [95], in the "Main findings" column, there is some typographical mistake: "-significant positive correlation between RT difference and ????????????...."

  • We thank the reviewer for pointing out this shortcoming. While in the latest word version of the manuscript the sentence is correctly displayed, in the PDF version the mentioned mistake occurs. The sentence should read as follows: “- significant positive correlation between RT difference and VO2 peak in ADHD“. We hope that it is now correctly displayed.

In the references, the journals of some articles are not presented with the abbreviation, maybe it can´t because there is not an abbreviation for these journals, but please, check it. I find them in lines: 618, 636, 644, 647, 660, 670, 678, 685, 691, 697, 741, 742, 747, 758, 885, 887, 903, 999, 1053, 1056, 1103. And maybe in others, I did not see.

  • Many thanks for your careful proofreading and for pointing out this limitation. We have edited the references and added journal abbreviations if possible.

  • In addition, please note that we have used the MDPI English Editing Service to improve the language in our manuscript. Unfortunately, after English editing not all of our changes are visible in “track changes” mode.

Reviewer 2 Report

The authors provide an interesting systematic review providing insight as to the use of fMRI for studies of acute exercise. The body of the manuscript is well written, but the abstract and introduction need additional refinement.

Abstract

Line 16- Change exercises to exercise.

Line 18- add a comma after improvements at remove the word yet.

Line 18- place a comma after techniques

Line 19- place a comma after (fMRI).

Line 21- remove the word on.

Line 23- databases is one word and remove the s from criterion

Line 26- remove the word their

Introduction

Line 41- Using the term rest to describe a break for physical activity is confusing.  Please reword this sentence perhaps as follows: In order to avoid periods of prolonged physical inactivity, it is advised to take show physical activity breaks. 

Methods

What protocol was followed for this systematic review? Why was Prisma, followed/not followed: http://prisma-statement.org/Protocols/

Why was the term physical activity and its derivatives not used in the search?

Results

Table 1 is helpful. I recommend referencing it more often.

Discussion

Line 342- It is unclear why the authors uses the term exercises as opposed to exercise throughout the manuscript.

Author Response

Abstract

Line 16- Change exercises to exercise.

  • We thank the reviewer for this hint and changed the sentence as suggested.
  • “The fact that a single bout of acute physical exercise has a positive impact on cognition is well-established in the literature, but the neural correlates which underlie these cognitive improvements, are not well understood.”

Line 18- add a comma after improvements at remove the word yet.

  • We are very thankful for this language advice. We used the MDPI language editing service to correct these shortcomings in the entire manuscript.

Line 18- place a comma after techniques

  • As suggested, we added a comma after ‘techniques’.
  • “Here, the use of neuroimaging techniques, such as functional magnetic resonance imaging (fMRI), offers great potential that has just started to be recognized.”

Line 19- place a comma after (fMRI).

  • We added the recommended comma.
  • “Here, the use of neuroimaging techniques, such as functional magnetic resonance imaging (fMRI), offers great potential that has just started to be recognized.”

Line 21- remove the word on.

  • We removed the word ‘on’ prior to cognition.
  • “This review aims at providing an overview of that studies that used fMRI to investigate the effects of acute physical exercises on cerebral hemodynamics and cognition.”

Line 23- databases is one word and remove the s from criterion

  • We thank the reviewer for pointing out this language mistakes. We corrected the sentences according to her/his suggestions.
  • “To this end, a systematic literature survey was conducted by two independent reviewers across five electronic databases.”

Line 26- remove the word their

  • We removed the word ‘their’ in the mentioned sentence.
  • “Although the findings of the reviewed studies suggest that acute physical exercise (e.g., cycling) leads to profound changes in functional brain activation, the small number of available studies and the great variability in the study protocols limits the conclusions that can be drawn with certainty.”

Introduction

Line 41- Using the term rest to describe a break for physical activity is confusing.  Please reword this sentence perhaps as follows: In order to avoid periods of prolonged physical inactivity, it is advised to take show physical activity breaks.

  • We are thankful for this language advice and edited the sentences as recommended by the reviewer.
  • “In order to avoid periods of prolonged physical inactivity (e.g., sedentism during office working day), it is advised to take physical activity breaks [23–25]”

Methods

What protocol was followed for this systematic review? Why was Prisma, followed/not followed: http://prisma-statement.org/Protocols/

  • Thank you again for your valuable comment. We revised our manuscript according to the recommendations provided in the PRISMA statement and added, in turn, a risk of bias assessment (see Figure 3).

Why was the term physical activity and its derivatives not used in the search?

  • We thank the reviewer for this important hint and added the terms “acute bout of physical activity” OR “single bout of physical activity” OR “acute physical activity” OR “physical activity break” to the renewed literature search. By using the renewed literature search 662 articles were identified in the five electronic databases from which 14 studies were considered as relevant (based on our inclusion and exclusion criteria’s). In comparison to the “old” literature search (i.e., 13 relevant studies), another relevant study was identified by using the “renewed” literature search (i.e., 14 relevant studies).

Results

Table 1 is helpful. I recommend referencing it more often.

  • We thank the reviewer for this hint and referenced Table 1 more often throughout the running text of the manuscript.

Discussion

Line 342- It is unclear why the authors uses the term exercises as opposed to exercise throughout the manuscript.

  • We are thankful for pointing out this shortcoming and to address this limitation, we revised the mentioned sentence and the entire manuscript carefully.

  • In addition, please note that we have used the MDPI English Editing Service to improve the language in our manuscript. Unfortunately, after English editing not all of our changes are visible in “track changes” mode.
